# Effect of Selenium Source and Level on Performance, Egg Quality, Egg Selenium Content, and Serum Biochemical Parameters in Laying Hens

**DOI:** 10.3390/foods9010068

**Published:** 2020-01-08

**Authors:** Hu Liu, Qifang Yu, Chengkun Fang, Sijia Chen, Xiaopeng Tang, Kolapo M. Ajuwon, Rejun Fang

**Affiliations:** 1 College of Animal Science and Technology, Hunan Agricultural University, No. 1 Nongda Road, Furong District, Changsha 410128, China; Tigerliu18@163.com (H.L.); yqqah@126.com (Q.Y.); clark_fang@163.com (C.F.); jysj2020@aliyun.com (S.C.); tangxiaopeng110@126.com (X.T.); 2 Hunan Co-Innovation Center of Animal Production Safety, Changsha 410128, China; 3 Department of Animal Sciences, Purdue University, West Lafayette, IN 47907-2054, USA; kajuwon@purdue.edu

**Keywords:** sodium selenite, selenium yeast, Roman laying hens, egg yolk selenium, laying rate

## Abstract

The objective of this study was to compare the effect of sodium selenite (SS) and selenium yeast (SY) on performance, egg quality, and selenium concentration in eggs and serum biochemical indices in laying hens. Seven hundred twenty healthy Roman laying hens (21 weeks old, 18 weeks in lay) with a similar laying rate (90.27% ± 1.05%) were randomly divided into 5 groups with 6 replicates of 24 hens each. Five diets were prepared as a 1+2×2 factorial arrangement with control and two sources of Se at two levels. Control diet (control) was prepared without adding exogenous selenium (analyzed basal Se content of 0.178 mg/kg). The other four diets were prepared with the control diet supplemented with SY or SS at 0.3 mg/kg (low; L) or 0.5 mg/kg (high; H) to give 5 diets designated as control, SY-L, SY-H, SS-L, and SS-H. The analyzed selenium content in the SY-L, SY-H, SS-L, and SS-H diets were 0.362, 0.572, 0.323, and 0.533 mg/kg respectively. The pre-trial period lasted 7 d, and the experimental period lasted 56 d (30 weeks old), during which the egg production, egg quality, and hen serum parameters were measured. Results showed that selenium source and level had no effect (*P* > 0.05) on average daily egg weight and feed conversion ratio (FCR). However, the laying rate was different at the L and H levels of supplementation, regardless of source, such that hens that were supplemented had a higher performance than that of the control, and the H level of supplementation lead to a higher laying rate than that of the L level (*P* < 0.05). There was a difference in average daily feed intake (ADFI) with an interaction in selenium source and level (*P* < 0.05), such that SS-L was higher than other selenium supplemented treatment or control. There were no significant differences in egg quality (*P* > 0.05); at the high level, SY had higher egg yolk selenium compared with SS. However, within SY, adding 0.5 mg/kg selenium led to higher egg yolk selenium than 0.3 mg/kg selenium (*P* < 0.05). Moreover, adding 0.3 mg/kg SY, 0.3 mg/kg, or 0.5 mg/kg SS to the basal diet had no significant effect on the selenium content in the egg (*P* > 0.05). There were no significant differences in serum biochemical indices among the five groups (*P* > 0.05). In conclusion, adding a high level of selenium in the diet of laying hens significantly increased egg production, and addition of a high level of selenium in the form of SY led to a higher deposition of selenium in the yolk than that of SS. These results indicate that adding 0.5 mg/kg of SY in the diet of laying hens would result in Se-enriched eggs.

## 1. Introduction

As an essential element for organisms, selenium (Se) is an important component of unusual amino acids in laying hens and includes selenocysteine and seleno-methionine. Selenium functions as a cofactor for the antioxidant function of enzymes like glutathione peroxidases and certain forms of thioredoxin reductase [1,2,3]. A deficiency of Se causes reduced fertility, placental retentions, and incidence of mastitis and metritis [3,4]. There are many places where the soil is deficient in Se around the world such as China, Northeast Asia, and Central Africa [4,5]. Hospital patients may present with Se deficiency symptoms when maintained with parenteral feeding without Se for a long period of time [6,7,8]. Rocha et al. (2016) found that supplementation with Se reduced neutropenic cases (leukemia/lymphomas (LL) and solid tumors (ST) in patients), reduced IgG and IgA levels in LL, and increased IgG and IgA levels in the ST group [9]. Freitas et al. (2014) also showed that prematurity and low birth weight could contribute to low blood selenium in premature infants. Selenium supplementation seems to minimize or prevent clinical complications caused by prematurity [10]. However, Freitas et al. (2017) confirmed selenium supplementation was not enough to reach the reference values during seven days of Se supplemented in patients with inflammation [11]. Therefore, it is necessary to increase Se content in common human foods, especially in foods produced in Se-deficient areas of the world. Selenium-enriched eggs can be produced by adding Se additives in hens’ diets. Asadi et al. (2017) reported that organic Se increased Se deposition in the egg and improved egg quality compared with the other sources of Se [12]. Some researchers indicated Se yeast might be a superior organic source of selenium compared to other selenite sources because of the better utilization and absorption by the animal [13,14,15,16]. Qu et al. [17] reported that adding selenium nanoparticles in feed of laying hens had no effect on egg weight, Haugh units, yolk color, eggshell strength, and eggshell thickness. However, Se deposition in eggs from laying hens fed with different selenium sources and levels has not been clarified. Selenium is an important trace element with a well-established antioxidant function and mechanism [1]. However, the influence of additional selenium supplementation on albumin (ALB), total protein (TP), and blood urea nitrogen (BUN) remains unclear. Therefore, the purpose of this study was to investigate the effects of supplementation of different selenium sources, in the form of sodium selenite (SS) and selenium yeast (SY), and levels (0.3 or 0.5 mg/kg) on performance, egg quality, egg selenium concentration, and serum biochemical indexes for laying hens. 

## 2. Materials and Methods

### 2.1. Experimental Design and Animal Assignment

This experiment was conducted on a commercial poultry farm (Yiyang city, Hunan province, China). All animal care procedures were approved by the Committee of Laboratory Animal Management and Animal Welfare of Hunan Agricultural University (Changsha city, Hunan province, China) and the Ethical Committee of Hunan Agricultural University (Ethics Approval number 201607-5). Seven hundred twenty Roman laying hens around 21 weeks old were selected with similar body size and laying rates. Hens were divided into 5 groups in a completely randomized design. Each group consisted of 6 replicates with 24 hens per replicate.

### 2.2. Animal Management 

Hens were housed at 4 birds per cage in a caged layer house. The trial lasted 56 d. Birds were fed 2 times per day at 05:00 and 13:00, and water and feed were provided ad libitum. The feeders and drinkers were checked every day. Feeding management and vaccination programs were concordant to common practice.

### 2.3. Experimental Diets 

A corn–soybean meal basal diet (Table 1) was supplemented with 0 (control diet), 0.3 mg/kg (low; L), and 0.5 mg/kg (high; H) Se from selenium yeast (SY, Se > 0.3%) or sodium selenite (SS, Se > 1%) respectively. All diets were formulated to meet nutrient requirements suggested by the National Research Council (NRC, 1994) [18].

## 3. Sampling and Analysis

### 3.1. Feed Sampling 

All diets were sampled and stored at 4 °C in refrigerators. Diet samples were analyzed in duplicate for Se using the method described in the Association of Official Analytical Chemists (AOAC) [19].

### 3.2. Measurements of Performance

Feed intake and egg production were recorded every day. The feed conversion ratio was estimated as kilograms of feed consumed per kilogram of eggs. At day 56 of the experiment, 6 eggs per replicate were randomly selected, and Se concentration, egg weight, Haugh units, and egg shell thickness were measured. Egg yolks were separated from whole eggs, dried at 65 °C for 12 h, and ground for Se analysis. Selenium content was determined by an inductively coupled plasma mass spectrometer (ICP-MS) according to Joaquim et al. [20].

### 3.3. Blood Sample Collection

Two hens in each replicate were randomly selected for blood sample collection at day 56. Blood samples were collected from the wing vein using a vacuum tube, kept at room temperature for 30 min, and then followed by centrifugation at 3000 r/min for 10 min. Serum was collected into a 1.5 mL tube and stored at −20 °C. The albumin assay kit, total protein assay kit, and blood urea nitrogen test kit were purchased from Nanjing Jiancheng Bioengineering Institute (Nanjing, China).

### 3.4. Statistical Analysis

All data were submitted to 2-way analyses of variance (ANOVA) using SPSS software (SPSS 21, SPSS Inc., Chicago, IL) to clarify the effects of dietary Se source, level, and their interaction. Tukey’s test was used to separate the means when the treatment difference was significant (*P* < 0.05). All data were expressed as the mean ± standard deviation. Statistical significance was considered at *P* ≤ 0.05.

## 4. Results

### 4.1. Performance Indices

As shown in Table 2, there were no differences with Se level and source for egg weight and the feed conversion. However, there was an interaction (*P* = 0.023) in ADFI between Se source and level. The hens fed a diet with 0.3 mg/kg SS (SS-L) had a greater (*P* < 0.05) ADFI than that of other treatments. The productivity of Se-supplemented laying hens was significantly higher than that of the basal diet, and the hens fed a diet with 0.5 mg/kg SY (SY-H) were highest. There was no treatment or interaction effect (*P* > 0.05) on soft or cracked eggs.

### 4.2. Egg Quality

As shown in Table 3, there was no significant difference from basal (control) diet and in egg shape index, eggshell thickness, eggshell strength, yolk weight, yolk index, albumen height, and Haugh unit among hens fed with either Se sources at either of the two levels, and there was no interaction effect (*P* > 0.05).

### 4.3. Selenium Concentrations in Egg Yolk 

The effect of adding different selenium sources and levels on Se concentration in egg yolk at day 56 is shown in Table 4 and Figure 1 and Figure 2. Compared with the control diet, the Se content in the egg yolk was significantly increased by adding supplemental selenium from both sources and at the two levels (*P* < 0.05). At the high level (0.5 mg/kg), SY significantly increased (*P* < 0.001) egg yolk Se accumulation compared to that of the two SS levels. Within the SY treatment, a high Se level led to a higher egg yolk Se compared to that of the low Se level (*P* = 0.009). These results showed that supplementation of different Se sources and levels had significant effects on increasing the Se content of eggs. The addition of 0.5 mg/kg SY led to the highest Se content. However, although still higher than the control diet, Se content was not different among treatments supplemented with 0.3 mg/kg SY, 0.3 mg/kg SS, or 0.5 mg/kg SS (*P* > 0.05). From Figure 1 and Figure 2, we conclude that the content of Se deposited in eggs was improved with increasing levels of additional SY or SS, respectively. The regression linear equations are: Y = 0.761X + 0.226 (R^2^ = 0.959, where Y represents the selenium content in the egg, and X represents the level of added selenium in the diets), and Y = 0.516X + 0.286 (R^2^ = 0.827) respectively. The quadratic regression equations are: Y = X^2^ + 0.716X + 0.2678 (R^2^ = 0.953), and Y = −1.07 × 2 + 1.020X + 0.268 (R^2^ = 0.904), respectively.

### 4.4. Serum Biochemistry

The effects of different Se sources and levels on serum biochemical indices are shown in Table 5. There were no significant differences in serum albumin, total protein, and urea nitrogen among all groups.

## 5. Discussion

### 5.1. Performance

The development and utilization of safe and efficient Se sources has been one of the major topics in animal nutrition and feed industry. Various kinds of Se products are widely used in animals such as yeast Se, nano-Se, amino acid Se, and so on. However, studies on the effect of supplementation of Se in the diet on performance of hens have been inconsistent. Gjorgovaka et al. [21] found that supplementation of organic Se in laying hens increased egg production percentages. Attia et al. [14] indicated that the feed conversion ratio decreased by Se supplementation (feed conversion ratio (FCR): 3.82 and 3.84 of sodium selenite and selenomethionine respectively) compared with hens fed the control diet (FCR: 4.52). However, some studies showed that performance was affected by neither Se source nor level supplementation [14,22,23]. Our results showed no significant differences in average daily egg weight and feed conversion ratio among the five diets, but there were significant differences in the laying rate, average daily feed intake, and soft or crashed egg rate.

### 5.2. Egg Quality

Patton [24] reported that supplementation of SS or SY had no effect on Haugh unit values in eggs compared with eggs from hens fed the basal diet. Payne et al. [25] found that the addition of SY to the diet decreased the rate of storage of eggs and increased the storage time of eggs. Our results agree with Attia et al. (2010) who found that adding Se in the diets had no significant effect on any traits of egg quality [14].

### 5.3. Selenium Content of Egg Yolk

Selenium transfer to the egg depends on its source and level in the diets. Urso et al. (2015) indicated that the content of Se in egg yolk increased when adding 0.15 to 0.3 mg/kg Se in the diet [26]. Surai et al. (2014) reported that the provision of organic Se in concentrations ranging from 0.3 to 0.5 mg/kg in the diet resulted in the accumulation of about 30% greater Se compared to that of the same concentrations of inorganic Se [27]. The results of this experiment agree with these previous reports. In this study, we also found significant quadratic relationships between Se supplementation level and Se content in the yolk. This relationship between diet selenium content and egg selenium disagreed with the previous studies. The possible reasons may be related to the strains of birds used, age, dietary composition, Se source, selenium level, and hen health conditions [28,29,30,31]. 

### 5.4. Serum Biochemical Indices

Albumin is one of the important protein sources synthesized in the liver, and its main functions include maintaining the plasma osmotic pressure, providing energy and repairing tissue, and as a carrier involved in the transport of nutrients to maintain the body tissue protein dynamic balance [32]. Yang et al. [33] showed no significant difference in total protein and albumin with the addition of 0.15, 0.30, 0.45, and 0.60 mg/kg sodium selenite. Zhang et al. [34] showed that there was no significant difference in serum urea nitrogen when sodium selenite was supplemented to Simmental steers at 0, 0.1, 0.2, and 0.3 mg/kg. In our study, we found that supplemental selenium did not affect blood albumin, total protein, and blood urea nitrogen, which agrees with these studies.

## 6. Conclusions

In conclusion, adding a high level of selenium in the diet significantly improved the egg production performance of laying hens, and SY supplementation was a more effective at increasing Se deposition in the yolk than SS. Specifically, adding 0.5 mg/kg SY in the diet of laying hens led to the production of Se-enriched eggs, which might be a valuable source of Se to support optimal human health.

## Reference

## Figures and Tables

**Figure 1 foods-09-00068-f001:**
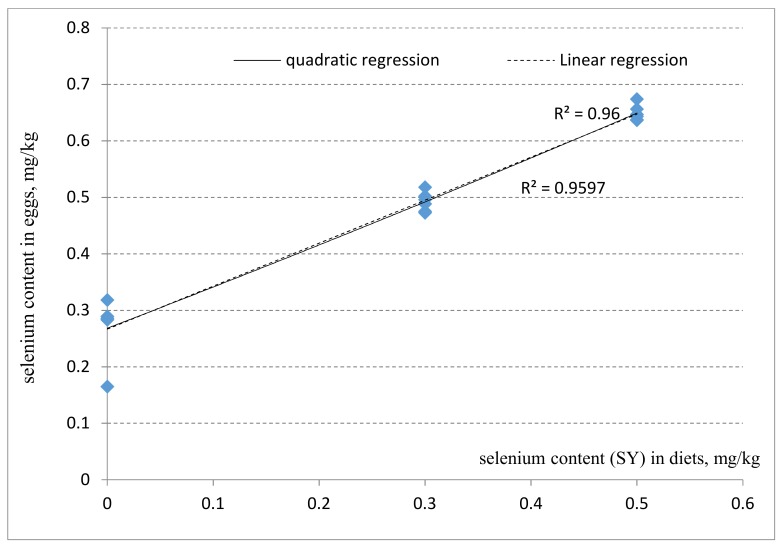
Linear and quadratic regression of egg selenium content (mg/kg) of laying hens on supplemental different level of yeast selenium (mg/kg).

**Figure 2 foods-09-00068-f002:**
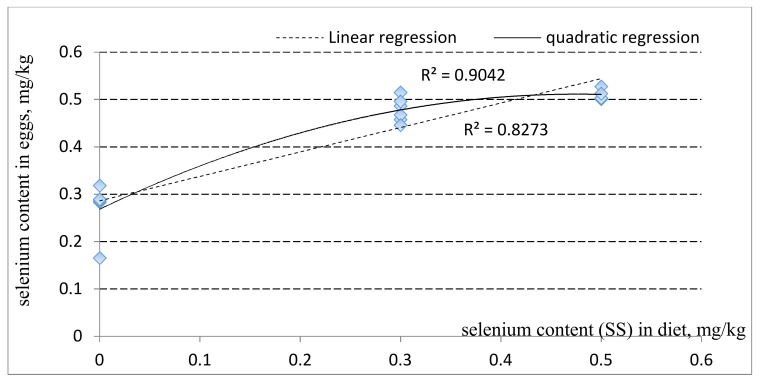
Linear and quadratic regression of egg selenium content (mg/kg) of laying hens on supplemental different levels of sodium selenite (mg/kg).

**Table 1 foods-09-00068-t001:** Composition and nutrients of the basal diet (air-dry basis).

Ingredients	Content (%)	Nutrition level	Content
Corn	61.0	ME, MJ/kg ^(2)^	11.12
Soybean meal	23.0	CP, %	15.90
Limestone	8.00	Ca, %	3.50
Rapeseed meal	3.00	AP, %	0.34
Soybean oil	1.00	Lys, %	0.84
Premix ^(1)^	4.00	Met, %	0.33

^(1)^ The premix provided the following per kg of diet: VA, 7715 IU; VD_3_, 2755 IU; VE, 8.8 IU; VK, 2.2 mg; VB_12_, 0.01 mg; VB_2_, 4.41 mg; VB_3_, 5.51 mg; VB, 0.55 mg; nicotinic acid, 19.8 mg; folic acid, 0.28 mg; Mn, 50 mg; Fe, 25 mg; Cu, 2.5 mg; Zn, 50 mg; and I, 1.0 mg. ^(2)^ Calculated according to NRC (1994).

**Table 2 foods-09-00068-t002:** Effects of dietary supplementation of different selenium sources and levels on production performance of laying hens.

Parameters	Basal Diet	SY-L	SY-H	SS-L	SS-H	*P-*Values		
						S	L	S × L
ADFI, g/d	114.58 ± 2.08 ^a^	114.73 ± 5.13 ^a^	114.39 ± 5.08 ^a^	116.05 ± 6.32 ^b^	114.43 ± 5.03 ^a^	0.106	0.438	0.023
AEW, g	55.47 ± 0.65	56.00 ± 2.68	55.47 ± 2.96	55.03 ± 2.48	55.22 ± 2.40	0.122	0.740	0.209
FCR	2.29 ± 0.04	2.24 ± 0.20	2.19 ± 0.14	2.30 ± 0.16	2.21 ± 0.09	0.075	0.716	0.366
Laying Rate, %	90.14 ± 0.99 ^a^	92.05 ± 6.84 ^b^	94.72 ± 5.06 ^c^	92.04 ± 5.23 ^b^	93.95 ± 3.71^c^	0.679	0.022	0.525
Soft or Cracked Eggs, %	0.63 ± 0.32 ^b^	0.61 ± 0.99 ^b^	0.36 ± 0.49 ^a^	1.27 ± 1.43 ^c^	0.67 ± 0.76 ^b^	0.191	0.081	0.118

Note: SY-L, 0.3 mg/kg selenium yeast; SY-H, 0.5 mg/kg selenium yeast; SS-L, 0.3 mg/kg sodium selenium; SS-H, 0.5 mg/kg sodium selenium; ADFI, average daily feed intake; AEW, average egg weight; FCR, feed conversion ratio; S, selenium sources; L, selenium levels; S × L, the interaction of selenium sources and levels. In the same row, means without letters or with the same superscripts are not significantly different (*P* > 0.05), while means with different superscripts mean significantly different (*P* < 0.05).

**Table 3 foods-09-00068-t003:** Effects of dietary supplementation of different selenium sources and levels on egg quality of laying hens.

Parameters	Basal Diet	SY-L	SY-H	SS-L	SS-H	*P*-Values		
						S	L	S × L
Egg shape index	1.30 ± 0.03	1.31 ± 0.10	1.29 ± 0.03	1.31 ± 0.06	1.30 ± 0.04	0.982	0.374	0.189
Eggshell thickness, mm	0.50 ± 0.30	0.48 ± 0.03	0.49 ± 0.04	0.50 ± 0.03	0.48 ± 0.02	0.629	0.669	0.143
Eggshell strength, kg/m^2^	5.27 ± 0.62	4.87 ± 0.87	4.89 ± 0.87	5.04 ± 1.02	4.94 ± 0.64	0.654	0.875	0.550
Yolk weight, g	15.60 ± 0.64	15.69 ± 0.54	15.84 ± 0.37	15.66 ± 0.85	15.74 ± 1.18	0.813	0.706	0.097
Yolk index	0.38 ± 0.04	0.39 ± 0.04	0.39 ± 0.04	0.41 ± 0.04	0.43 ± 0.02	0.281	0.144	0.102
Albumen height, mm	5.65 ± 0.36	5.88 ± 0.90	5.35 ± 0.81	5.53 ± 0.56	5.24 ± 1.12	0.058	0.437	0.539
Haugh unit	75.01 ± 2.87	75.62 ± 4.67	75.52 ± 4.02	75.34 ± 3.35	75.54 ± 3.97	0.913	0.973	0.973

Note: SY-L, 0.3 mg/kg selenium yeast; SY-H, 0.5 mg/kg selenium yeast; SS-L, 0.3 mg/kg sodium selenium; SS-H, 0.5 mg/kg sodium selenium; S, selenium sources; L, selenium levels; S × L, the interaction of selenium sources and levels. In the same row, means without letters or with the same superscripts are not significantly different (*P* > 0.05).

**Table 4 foods-09-00068-t004:** Effects of dietary supplementation of different selenium sources and levels on egg selenium content of laying hens.

Parameters.	Basal Diet	SY-L	SY-H	SS-L	SS-H	*P*-Values		
						S	L	S × L
Se content, mg/kg	0.2683 ± 0.0593 ^A^	0.4920 ± 0.0171 ^B^	0.6491 ± 0.0142 ^C^	0.4780 ± 0.0257 ^B^	0.5107 ± 0.0120 ^B^	0.009	<0.001	<0.001

Note: SY-L, 0.3 mg/kg selenium yeast; SY-H, 0.5 mg/kg selenium yeast; SS-L, 0.3 mg/kg sodium selenium; SS-H, 0.5 mg/kg sodium selenium; S, selenium sources; L, selenium levels; S × L, the interaction of selenium sources and levels. In the same row, means without letters or with the same superscripts are not significantly different (*P* > 0.05), while means with different superscripts mean significantly different (*P* < 0.05).

**Table 5 foods-09-00068-t005:** Effects of dietary supplementation of different selenium sources and levels on serum biochemical indexes of laying hens.

Parameters	Basal Diet	SY-L	SY-H	SS-L	SS-H	*P*-Values		
						S	L	S × L
ALB, g/L	24.58 ± 1.74	25.80 ± 4.67	26.52 ± 4.28	24.08 ± 2.95	25.90 ± 2.38	0.453	0.410	0.712
TP, mg/ml	54.71 ± 3.13	58.17 ± 7.75	59.75 ± 5.71	55.56 ± 3.89	57.45 ± 3.32	0.302	0.144	0.450
BUN, mmol/L	5.32 ± 1.12	5.55 ± 0.66	5.75 ± 0.29	5.42 ± 1.24	5.35 ± 1.10	0.589	0.818	0.931

Note: SY-L, 0.3 mg/kg selenium yeast; SY-H, 0.5 mg/kg selenium yeast; SS-L, 0.3 mg/kg sodium selenium; SS-H, 0.5 mg/kg sodium selenium; ALB, albumin; TP, total protein; BUN, blood urea nitrogen; S, selenium sources; L, selenium levels; S × L, the interaction of selenium sources and levels.

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
