# Peer review of "Effect of Selenium Source and Level on Performance, Egg Quality, Egg Selenium Content, and Serum Biochemical Parameters in Laying Hens"

_foods, 2020, doi:10.3390/foods9010068_

Round 1

Reviewer 1 Report

General comments

The article deals with the effects of Selenium supplementation in the diet of laying hens - quantity and source - on the respective performance, egg properties, and serum biochemical parameters.

The study is not completely new, but the focus on Se content in eggs is sufficiently original and conveys useful information.

The research design is quite satisfying, and the results are clear.

However, language and style are rather poor, and few important statements are unclear or unsupported, as well as few results are underdocumented.

Finally, the format of the manuscript does not comply with the Journal's template.

Specific comments

Due to many specific comments, they are made available in the herein enclosed document.

I fixed only part of the language and style errors. Authors are urged to revise the manuscript, possibly with the support of a native English speaker.

Authors are as well invited to consider all my comments, respond where applicable, and react accordingly.

Author Response

Response to Reviewers

Manuscript ID: foods-671880

Title: "Effect of Selenium Source and Level on Performance, Egg Quality, Egg Selenium Content and Serum Biochemical Parameters in Laying Hens".

Correspondence author: Rejun Fang, Email: [email protected]

Dear editor,

Thank you very much for your attention and the reviewer’s evaluation and comments on our manuscript entitled " Effect of Selenium Source and Level on Performance, Egg Quality, Egg Selenium Content and Serum Biochemical Parameters in Laying Hens" (Manuscript ID: foods-671880).

Those comments are all valuable and very helpful for revising and improve our paper, as well as the important guiding significance to our researches. We have studied comments carefully and have revised the manuscript according to your kind advices and reviewer’s detailed suggestions. Revised portion are marked in the paper. Enclosed please find the responses to the reviewers. We revised the manuscripts by a native English speaker who name is Kola and worked in Purdue University. Also, according to the rules of foods, we confirm Hu Liu is the only one first author.

We sincerely hope this manuscript will be finally acceptable to be published on foods.

Thank you very much for all your help and looking forward to hearing from you soon.

Best regards

Sincerely yours

Dr. Hu Liu

[email protected]

Please find the following Response to the comments of reviewers:

Response to the reviewers’ comments

Reviewer: 1 

We revised language by Kola who working in Purdue University again.

In line 23, the unit of laying rate not showed.

Response: Thank you. No unit on laying rate. It is the ration of number of egg and laying hens on the experiments period in laying hens production included in this manuscript.

In line 26 where does this Se content come from? Does it refer to control?

Response: Thanks for your attention. The Se content of 0.178 mg/kg is analyzed and refers to control Se concentration. The selenium comes from the ingredients of diets. In our experiments, major comes from corn, soybean meal, rapeseed meal.

In line 31 “ … which the egg production and, egg quality and hen serum parameters were measured.” Deleted the first “and”.

Response: Thanks. We deleted it in the manuscript.

In line 36 “… and the H level of supplementation was higher than the L level (P < 0.05)” changed into “… and the H level of supplementation led to higher laying rate than the L level (P < 0.05)”

Response: Thank you for your attention. We changed it and you can read in line 35.

In line 37, all abbreviations must be defined in the Abstract, too.

Response: Thanks a lot. We redefined the abbreviations in the abstract.

In line 47, the key words of sodium selenite (SS), selenium yeast (SY) deleted the abbreviations.

Response: Thank you. We deleted it.

In line 54, Likely, scavening of oxidants, moreover the text is unclear and should be rephrased.

Response: Thank you for your suggestions. So many previous studies showed the antioxidant capability of selenium because it is the key co-factor mineral on glutathione peroxidases. In our views, we thought it is suitable to shown scavenging of oxidants in our manuscripts. Also, Reyes et al. (2019) reported that supplementation selenomethionine can scavening of oxidants.

Reyes, L., Bishop, D. P., Hawkins, C. L., & Rayner, B. S. 2019. Assessing the Efficacy of Dietary Selenomethionine Supplementation in the Setting of Cardiac Ischemia/Reperfusion Injury. Antioxidants, 8(11), 546. doi:10.3390/antiox8110546

In line 59, Too many references together. Either reduce to maximum three, or briefly explain contents for groups of maximum three.

Response: Thank you. We changed it and modified it in detail in line 58.

In line 67, “However, the Se deposition in eggs from laying hens fed different selenium sources and level is not clarified”. Changed into “… fed with ….”.

Response: Thank you. We changed it in line 75.

In line 68, “Selenium as an important element trace, the mechanism of antioxidant is very popular.” is unclear statement, to be rephrased.

Response: Thank you. We changed into “Selenium as an important element trace which consisted of glutathione peroxidases, the mechanism of antioxidant is very popular (1)”.

Rotruck J T, Pope A L, Ganther H E, Swanson A B, Hafeman D G, & Hoekstra W G. 2010. Selenium: biochemical role as a component of glutathione peroxidase. Science, 179:588-590.

In line 70, “ALB, TP and BUN” should define acronyms.

Response: Thank you for a lot. We define abbreviations and changed into “albumin (ALB), total protein (TP) and blood urea nitrogen (BUN)” in line 78.

In line 88, “Feeding management and vaccination programs were concordant to common practice.” should be cited a reference.

Response: Thank you for a lot. Actually, farm has its rules to feed laying hens. Moreover, different farms have different feeding management and vaccination programs. Also, all animals care procedures were approved by the committee of Laboratory Animal Management and Animal Welfare of Hunan Agricultural University and Ethical Committee of Hunan Agricultural University (ethics approval number 201607-5). Unfortunately, there is no reference to define how to management and vaccination.

In line 98, define the abbreviation “AOAC”.

Response: Thank you. AOAC is the “Association of Official Analytical Chemists” and we define it.

In line 100, “Feed intake and egg production were recorded per day” changed into “Feed intake and egg production were recorded every day”

Response: Thank you. We changed it.

In line 122, Why mention of "Soft or cracked eggs", where SY-H performed better? Is it relevant?

Response: Thanks. The soft or cracked eggs are an important index in laying hens performance. In our results, we founded that SY-H performed better but no difference among 5 treatments (P>0.05). Moreover, previous studies showed no different in selenium sources and levels (Han et al., 2017; Delezie et al, 2014) which in consistent with our results. However, the major reasons led to soft or crashed eggs may the content of calcium, phosphorus in diets and the health conditions of laying hens.

Han, X. J., Qin, P., Li, W. X., Ma, Q. G., Ji, C., Zhang, J. Y., & Zhao, L. H. 2017. Effect of sodium selenite and selenium yeast on performance, egg quality, antioxidant capacity, and selenium deposition of laying hens. Poultry Science, 96(11), 3973–3980. doi:10.3382/ps/pex216

Delezie, E., Rovers, M., Van der Aa, A., Ruttens, A., Wittocx, S., & Segers, L. 2014. Comparing responses to different selenium sources and dosages in laying hens. Poultry Science, 93(12), 3083–3090. doi:10.3382/ps.2014-04301

In line 125, “The hens fed diet with 0.3 mg/kg SS (SS-L) had greater (P< 0.05) feed intake than other treatments.” should be changed into “The hens fed with diet with 0.3 mg/kg SS (SS-L) had greater (P< 0.05) feed intake than other treatments.”

Response: Thank you. We changed it.

In line 129, No difference as well with regard to basal (control) diet - this information should be added.

Response: Thank you a lot. We added it in line 145.

In line 145, “we concluded that the content of Se increased with the increase of the addition level adding different levels of SY or SS” was unclear, to be reworded.

Response: Thank you very much. We checked it and revised into “we concluded that the content of Se deposited in eggs was improved with the increase of the addition level adding different levels of SY or SS, respectively. ”

In line 145, the regression equations with so few data, using quadratic regression might have little significance. Moreover, linear would be enough, based on Figures 1 and 2, or linear for Figure 1 and quadratic for Figure 2. Adding more regressions could generate confusion and add little information.

Response: Thank you for your attention. The data comes from 6 replicate and each replicate texted 2 eggs. But, we get the mean data of each replicate. In our views, there are enough to get the linear and quadratic regression and it is necessary to show the quadratic regression. Moreover, to produce selenium-rich eggs should be consider the environment effects and the efficiency deposit in eggs. So we should found the appropriate concentration on laying hens diets. In our further study, we will focus on the selenium cycle on laying hens-soil-plant (tea tree).

In 146, Equations should be numbered, as well as Y and X variables should be explained.

Response: Thank you for your attention. The Y means the selenium content in eggs, and X means the selenium content in diets. We were explained the Y and X in the equations.

In line 158, “However, the effect of supplementation of Se in diet on performance of hen are inconsistent.” changed into “However, studies on the effect of ….”

Response: Thank you. We changed into “. However, studies on the effect of supplementation of Se in diet on performance of hen were inconsistent.” and showed in line 175.

In line 163, the sentence of “Our results showed that no significant differences in average daily egg weight, feed conversion ration among the five diets but there were significant differences in the laying rate, average daily feed intake and broken rate and dirty egg rate.” changed into “Our results showed no significant differences in average daily egg weight, feed conversion ratio among the five diets but there were significant differences in the laying rate, average daily feed intake and soft or crashed egg rate.”

Response: Thank you. We changed it in line 84.

In line 160, Unclear: dod Se supplementation lead to improvement of feed conversion ratio, in comparison to control diet, or not?

Response: Thank you for your suggestions. We checked and changed into “Attia et al. (14) indicated that feed conversion ratio decreased by Se supplementation (FCR: 3.82 and 3.84 of sodium selenite and selenomethionine respectively) compared with hens fed the control diet (FCR: 4.52)”

In line 169, Define the meaning of the rate of storage and the storage time.

Response: Thank you. The rate of storage and the storage time refer to the shelf life. Payne et al. (2005) showed the egg was stored at 22.2℃ or 7.2 ℃ and supplemented with selenium were increased the antioxidant capability in eggs. Moreover, the Haugh unit was the index to evaluate the rate of storage and the time in laying hens production.

In line 181, Explain why and to which extent the result disagree.

Response: Thank you for your attention. We explained the reasons in line 182 to 184. Many factor influenced the relationship between diet selenium and egg selenium. But there is common that the egg deposited selenium were increased by the diets selenium supplemented concentration.

Reviewer 2 Report

This study provides additional information regarding the effects of selenium supplementation of hens' diet on egg parameters.  The study confirms results found in some of the literature.

My main concern is some confusion regarding interpretation of some of the results:

Table 2, "soft or cracked eggs, %": although the S, L, and SxL values are >0.05, superscript letters indicate significant differences among some of the treatments.  Also, this parameter is not reported in the results section.

lines 138-139 and Table 4: from the table it appears that only the SY-H treatment has a significantly different value than the other three treatments.  This is inconsistent with what is reported in lines 138-139.

lines 142-143: as above, the results in table 4 show a significant effect of all treatments.

Check how these results are reported in the abstract (e.g., lines 39 and 40)

line 165: is there any "dirty egg" data in this study?

lines 174 and 176: check the sources cited; they do not correspond with the reference list #26 and #27.

Table 1: the "nutritional level" column is confusing as is the "Content" column on the far right.

Table 2: I suggest including the word "selenium" in the title; as discussed above, the superscript letters in the "soft or cracked eggs, %" row are not addressed or indicated by the p-values.

Table 2, line 330: should be "row" rather than "column": "In the same row..."

Table 3: I suggest including the word "selenium" in the title

Table 3, line 339: should be "row" rather than "column": "In the same row..."

Table 4: I suggest including the word "selenium" in the title

Table 4, line 348: should be "row" rather than "column": "In the same row..."

Table 4: capitalize superscript letters

Table 5: I suggest including the word "selenium" in the title

Figure 1: include the R squared values

Figure 2: include the R squared values

Line 114: Statistical Analysis: include regression analysis in this section.

Some comments on English usage and writing style:

line 31: remove "and"

line 34: should be "levels"

line 37: "There was a difference..."; define "ADFI" abbreviation

line 42: "...among the 5 groups..."

line 51: "organisms"

line 53: remove "has"

line 55: add a period and capitalize A:  "(1-3). A"

lines 51 - 55: reword this sentence; it would be better as two sentences.

line 59: remove "the"

line 61: add a period after et al.

line 64: "...absorption by the animal..."

line 68: "trace element"

line 70: define abbreviations (ALB, TP, BUN)

line 77: "animal care" (rather than animals)

line 81: "21-weeks-old"

line 86: "...in caged layer house."

line 106: add a period after et al.

line 123: "As shown in table 2,..."

line 127: "...productivity of Se-supplemented laying hens..."

line 131: "yolk weight" is listed twice

line 132: "...hens fed with either  Se source, Se ..."

line 137: "...content in egg yolk was..."

line 152: "...and urea nitrogen among all groups."

line 155: "sources"

line 158: "hens"

line 162: "some studies show that..."

line 163: "...showed no significant differences..."

line 164: "feed conversion ratio among the five diets..."

line 167 "...supplementation of SS or SY..."

lines 171 - 172: be specific about which studies you are referring to and how your results agree with those specific studies.  Are you referring to all the studies mentioned in the paragraph?

line 177: "...in the diet resulted in accumulation of about 30%..."

lines 179 and 182: be specific about which studies you are referring to and how your results differ or agree.

lines 186 - 187: "...liver, and its main functions include maintaining..."

line 188: "...tissue, and as a carrier..."

line 193: "...we found that supplemental"

line 197: conclusion

line 200: health

line 208: "...any nature or kind in..."

Author Response

Response to Reviewers

Manuscript ID: foods-671880

Title: "Effect of Selenium Source and Level on Performance, Egg Quality, Egg Selenium Content and Serum Biochemical Parameters in Laying Hens".

Correspondence author: Rejun Fang, Email: [email protected]

Dear editor,

Thank you very much for your attention and the reviewer’s evaluation and comments on our manuscript entitled " Effect of Selenium Source and Level on Performance, Egg Quality, Egg Selenium Content and Serum Biochemical Parameters in Laying Hens" (Manuscript ID: foods-671880).

Those comments are all valuable and very helpful for revising and improve our paper, as well as the important guiding significance to our researches. We have studied comments carefully and have revised the manuscript according to your kind advices and reviewer’s detailed suggestions. Revised portion are marked in the paper. Enclosed please find the responses to the reviewers. We revised the manuscripts by a native English speaker who name is Kola and worked in Purdue University. Also, according to the rules of foods, we confirm Hu Liu is the only one first author.

We sincerely hope this manuscript will be finally acceptable to be published on foods.

Thank you very much for all your help and looking forward to hearing from you soon.

Best regards

Sincerely yours

Dr. Hu Liu

[email protected]

Please find the following Response to the comments of reviewers:

Response to the reviewers’ comments

Reviewer: 2:

We revised language by Kola who working in Purdue University again.

In Table 2, "soft or cracked eggs, %": although the S, L, and SxL values are >0.05, superscript letters indicate significant differences among some of the treatments. Also, this parameter is not reported in the results section.

Response: Thank you for your attention. We added “No difference (P> 0.05) on soft or cracked eggs on selenium sources, levels and its interactions.” in line 142.

In lines 138-139 and Table 4: from the table it appears that only the SY-H treatment has a significantly different value than the other three treatments. This is inconsistent with what is reported in lines 138-139.

Response: Thank you very much. We checked it carefully and revised into “At the high level (0.5mg/kg), SY significantly increased…”

In lines 142-143: as above, the results in table 4 show a significant effect of all treatments.

Response: Thank you. We checked the data and modified it into “The addition of 0.5 mg/kg SY was the highest but adding 0.3mg/kg SY, 0.3 mg/kg or 0.5 mg/kg SS in basal diet had no significant effect on the selenium content in egg (P> 0.05).”

Check how these results are reported in the abstract (e.g., lines 39 and 40)

Response: Thanks. We checked it and showed in line 39 to 43.

In line 165: is there any "dirty egg" data in this study?

Response: Thank you very much. We made a mistake in our manuscript. There is no dirty egg data in our study. So we deleted it. Also, we changed into “… intake and soft or crashed egg rate.” in line 184.

In lines 174 and 176: check the sources cited; they do not correspond with the reference list #26 and #27.

Response: Thank you very much. We checked it carefully and changed it.

Table 1: the "nutritional level" column is confusing as is the "Content" column on the far right.

Response: Thank you very much. The nutritional levels were meets the National Research Council (NRC, 1994) recommendations requirements on laying hens. Also, it was tested on the laboratory besides ME. Also the CP, Ca, AP, Lys, and Met were from the ingredients.

Table 2: I suggest including the word "selenium" in the title; as discussed above, the superscript letters in the "soft or cracked eggs, %" row are not addressed or indicated by the p-values.

Response: Thank you very much. We added “selenium” in the title. Also, in the “soft or cracked egg, %” rows, there is no difference among treatments. We defined P≤0.05 as significance.

Table 2, line 330: should be "row" rather than "column": "In the same row..."

Response: Thank you very much. We checked it carefully and revised it.

Table 3: I suggest including the word "selenium" in the title

Response: Thank you very much. We added “selenium” in the title.

Table 3, line 339: should be "row" rather than "column": "In the same row..."

Response: Thank you very much. We checked it carefully and revised it.

Table 4: I suggest including the word "selenium" in the title

Response: Thank you very much. We added “selenium” in the title.

Table 4, line 348: should be "row" rather than "column": "In the same row..."

Response: Thank you very much. We checked it carefully and revised it.

Table 4: capitalize superscript letters

Response: Thank you very much. We checked it carefully and revised it.

Table 5: I suggest including the word "selenium" in the title

Response: Thank you very much. We added “selenium” in the title.

Figure 1: include the R squared values

Response: Thank you. The regression showed in the results and we thought not necessary to show the R2 in figure 1.

Figure 2: include the R squared values

Response: Thank you. The regression showed in the results and we thought not necessary to show the R2 in figure 1.

Line 114: Statistical Analysis: include regression analysis in this section.

Response: Thank you. We added into the regression analysis in the statistical analysis.

Some comments on English usage and writing style:

In line 31: remove "and"

Response: Thanks for your attention. We removed it and showed in line 31.

In line 34: should be "levels"

Response: Thanks a lot. We changed it

In line 37: "There was a difference..."; define "ADFI" abbreviation

Response: Thanks a lot. We define “ADFI” is the average daily feed intake and changed in line 37.

In line 42: "...among the 5 groups..."

Response: Thank you for your suggestions. We changed followed by your suggestions.

In line 51: "organisms"

Response: Thank you for your advices. We checked it and changed.

In line 53: remove "has"

Response: Thank you. We removed it in line 54.

In line 55: add a period and capitalize A: "(1-3). A"

Response: Thanks. We modified it.

In lines 51 - 55: reword this sentence; it would be better as two sentences.

Response: Thanks. We changed into “As an essential element for organisms, selenium (Se) is an important component of unusual amino acids in laying hens including selenocysteine and seleno-methionine, and functions as a co-factor for the reduction of antioxidant which related on enzymes like glutathione peroxidases and certain forms of thioredoxin in reductase (1-3). A deficiency of Se causes reduced fertility, placental retentions, and the incidence of mastitis and metritis (3; 4)”

In line 59: remove "the"

Response: Thank you. We remove “the” in line 59.

In line 61: add a period after et al.

Response: Thanks. We added “2017” after et al.

In line 64: "...absorption by the animal..."

Response: thank you. However, we not sure what is the suggestion your give? Please more clarify.

In line 68: "trace element"

Response: Thank you for your suggestions. We changed followed by your suggestions.

In line 70: define abbreviations (ALB, TP, BUN)

Response: Thank you. We define abbreviations and changed into “albumin (ALB), total protein (TP) and blood urea nitrogen (BUN)” in line 81.

In line 77: "animal care" (rather than animals)

Response: Thank you for your attention. We modified it.

In line 81: "21-weeks-old"

Response: Thank you. We changed it.

In line 86: "...in a caged layer house."

Response: Thank you. We changed it.

In line 106: add a period after et al.

Response: Thank you. We changed it.

In line 123: "As shown in table 2,..."

Response: Thank you. We changed it.

In line 127: "...productivity of Se-supplemented laying hens..."

Response: Thank you. We changed it.

In line 131: "yolk weight" is listed twice

Response: Thank you. We changed it.

In line 132: "...hens fed with either Se source, Se ..."

Response: Thank you. We changed followed by your suggestion.

In line 137: "...content in egg yolk was..."

Response: Thank you. We modified it in line 152.

In line 152: "...and urea nitrogen among all groups."

Response: Thank you. We modified it into “…and urea nitrogen among all groups” in line 170.

In line 155: "sources"

Response: Thank you. We adding “s” and changed into “sources”

In line 158: "hens"

Response: Thanks. We adding “s” and changed into “hens”

In line 162: "some studies show that..."

Response: Thanks. The sentence of “… some study shows that” changed into “some studies show that …” which followed by your advices.

In line 163: "...showed no significant differences..."

Response: Thanks. We deleted “that” in line 182.

In line 164: "feed conversion ratio among the five diets..."

Response: thanks, we changed “5” into “five” which followed by your suggestion.

In line 167 "...supplementation of SS or SY..."

Response: thanks. We changed into “…supplementation of SS or SY...”.

In lines 171 - 172: be specific about which studies you are referring to and how your results agree with those specific studies. Are you referring to all the studies mentioned in the paragraph?

Response: Thanks. We changed it into “Our results in agree with Attia et al. (2010) found that adding Se in the diets had no significant effect on any traits of egg quality(14)”.

In line 177: "...in the diet resulted in accumulation of about 30%..."

Response: Thank you for a lot. We changed it followed by your advices.

In lines 179 and 182: be specific about which studies you are referring to and how your results differ or agree.

In lines 186 - 187: "...liver, and its main functions include maintaining..."

Response: Thank you for a lot. We modified it in line 204.

In line 188: "...tissue, and as a carrier..."

Response: Thank you very much. We added “as” in line 206.

In line 193: "...we found that supplemental"

Response: Thank you very much. We revised it followed by your suggestion.

In line 197: conclusion

Response: Thank you very much. We deleted “s” in the word “conclusions” in line 215.

In line 200: health

Response: Thank you very much. We deleted “y” in the word “healthy” in line 218.

In line 208: "...any nature or kind in..."

Response: Thank you a lot. We checked it and modified.

Round 2

Reviewer 1 Report

The Authors improved the manuscript even beyond my expectations. All my comments were properly responded, clarifying every point I raised, and the manuscript was properly amended.

The improvements in style and language were very careful and effective. Possible, occasional and minor issues could be fixed at the Proofs stage.

All statements are now supported and the experiments are reproducibile by other scholars. The article conveys useful information for both scientists and practitioners.